# Clinical Characteristics of TZAP (ZBTB48) in Hepatocellular Carcinomas from Tissue, Cell Line, and TCGA

**DOI:** 10.3390/medicina58121778

**Published:** 2022-12-02

**Authors:** Soo-Jung Jung, So-Hyun Kil, Hye Won Lee, Tae In Park, Yun-Han Lee, Jongwan Kim, Jae-Ho Lee

**Affiliations:** 1Department of Anatomy, School of Medicine, Keimyung University, 1095 Dalgubeol-daero, Daegu 42601, Republic of Korea; 2Department of Anesthesiology and Pain Medicine, CHA University, CHA Gumi Medical Center, Gumi 39295, Republic of Korea; 3Department of Pathology, Keimyung University School of Medicine, Daegu 42601, Republic of Korea; 4Department of Pathology, Kyungpook National University School of Medicine, Daegu 42601, Republic of Korea; 5Department of Molecular Medicine, School of Medicine, Keimyung University, Daegu 42601, Republic of Korea; 6Department of Biomedical Laboratory Science, Dong-Eui Institute of Technology, 54 Yangji-ro, Busan 47230, Republic of Korea

**Keywords:** TZAP, telomere, hepatocellular carcinoma, ZBTB48

## Abstract

*Background and Objectives*: ZBTB48 is a telomere-related protein that has been renamed telomeric zinc finger-associated protein (TZAP). It favorably binds to elongated telomeres to regulate their appropriate length. However, TZAP expression has not been investigated in hepatocellular carcinomas (HCC). *Materials and Methods*: The clinical significance of TZAP expression in 72 HCC was investigated. Additionally, its findings were supported by open big data and cancer cell lines. *Results*: TZAP expression level was not associated with the clinical parameters of HCC. TZAP expression induced a poorer survival result (overall survival, *p* = 0.020; disease-free survival, *p* = 0.012). TCGA data showed TZAP expression was more frequently found in HCCs with hepatitis C infection (*p* = 0.023). However, TCGA data revealed that TZAP expression did not predict HCC prognosis. In a cell line study, TZAP inhibition via siRNA suppressed PLC/PRF/5 cell growth; however, cell viability was increased in HepG2 cells. *Conclusions*: We presented the clinical and prognostic values of TZAP expression in HCC tissues and cancer cell lines. Additionally, the TCGA results also revealed a significant role for TZAP expression. TZAP expression may involve HCC progression and its prognosis.

## 1. Introduction

Telomeres have TTAGGG repeat sequences, and they are nucleoprotein structures that cap each end of chromosomes [1,2]. Telomeres in normal somatic cells are a typical length (5~15 kilobases), and their length was reduced by about 30–200 base pairs during cell division. In stem cells, a shortened telomere is counteracted by an alternative lengthening mechanism or the reverse transcriptase telomerase [3]. Cancer cells acquire telomere length maintenance mechanisms for replicative immortality, explaining the importance of these structures in oncology [4,5].Telomere elongation is thus induced by telomerase and alternative telomere lengthening mechanisms.

Recent studies showed a new mechanism by which elongated telomeres are changed to their typical length by rapid telomere shortening [6,7]. It was performed by ZBTB48, renamed telomeric zinc finger-associated protein (TZAP). Previous studies suggested that high expression of TZAP might induce progressive telomere shortening [8,9]. TZAP binds to TTAGGG telomere repeats directly, and it replaces telomeric repeat-binding factor 2 (TRF2), acting as the telomere capping factor [6]. Then, TZAP prevents excessive elongation of telomeres by inducing the initiation of telomere trimming. TZAP localizes to chromosome 1p36, and this region is commonly rearranged or deleted in human tumors [10,11,12]. TZAP may play an important role in cancer pathogenesis, but genetic research on it has been limited. Its genetic alteration was not common, being present in less than 5% of all kinds of cancers [13]. Relatively, its frequency was higher in adrenocortical carcinoma, cervical adenocarcinoma, and mature B-cell neoplasms. Additionally, TZAP expression and its prognosis were different according to cancer type. Interestingly, TZAP expression across all cancer samples as a whole was related to a better survival result. However, detailed studies about the clinical and prognostic characteristics of TZAP expression have not been fully conducted in each cancer type. 

Hepatocellular carcinoma (HCC) is the most common type of cancer in humans and the leading cause of death in Korea [14,15]. The major risk factors for HCC include liver cirrhosis, hepatitis B or C virus infection, alcohol abuse, and genetic alterations. Among many genetic factors, mutations in telomere-related genes, such as telomerase reverse transcriptase (TERT), were reported to occur frequently in approximately half of HCC patients [16]. Therefore, abnormalities in TERT expression or promoter mutations have been studied in HCC, and these data presented a comprehensive understanding of the entire scope of telomere biology in HCC [17,18]. Our findings, along with previous research, demonstrated the clinicopathological and prognostic importance of telomeres and TERT in HCC [16,17,18]. Additionally, we previously studied TZAP expression in colorectal cancer (CRC) via patient tissues, cell lines, and big data such as The Cancer Genome Atlas (TCGA) [19]. Additionally, clinical and prognostic values of TZAP expression and its associations with other genes were investigated. These studies suggested that telomere regulation had the potential to serve as a prognostic marker for cancer. 

Here, we examined TZAP expression in patients with HCC by determining their clinicopathological and prognostic characteristics. Then, the results were also investigated using publicly available big data and cancer cell lines. This comprehensive approach will provide molecular mechanisms of TZAP expression in carcinogenesis, especially hepatic carcinogenesis.

## 2. Materials and Methods

### 2.1. Patients and Tissue Samples

A total of 162 patients who underwent HCC surgery at the Kyungpook National University Hospital (KNUH) from January 2005 to December 2010 were enrolled. The institutional review board gave approval (KNUH-2014-04-056-001). Thirty-one samples were of poor quality, and 59 samples were excluded from the study due to an insufficient amount of RNA. Therefore, 72 samples covered the criteria for this study.

The clinicopathological data of each patient were evaluated, and patients with other malignancies were excluded. Additionally, preoperative therapies such as chemotherapy, radiofrequency ablation, and transarterial chemoembolization were excluded from this study. 

The tumor and paired non-malignant liver tissue were fixed with formalin and embedded in paraffin. Representative lesions were selected in paraffin blocks, and they were cored manually with a 3.0 mm diameter cylindrical device. Then, DNA and RNA extractions were performed as previously described [18,19]. Their quantity and quality were checked by NanoDrop 1000 (Thermo Fisher Scientific, Inc., Pittsburgh, PA, USA).

### 2.2. DNA Isolation and Telomere Length Analysis

We isolated DNA from samples using QIAamp DNA mini kits (Qiagen, Inc., Valencia, CA, USA). Telomere length was analyzed using qPCR with specific primers for telomere (T) and β-globin (S), based on previous study [18]. The sequences of telomeres (Bioneer, Daejeon, Republic of Korea) are as follows: Forward ‘CGG TTT GTT TGG GTT TGG GTT TGG GTT TGG GTT TGG GTT’ and reverse ‘GGC TTG CCT TAC CCT TAC CCT TAC CCT TAC CCT TAC CCT’. qPCR was performed by a LightCycler 480 II system (Roche Diagnostics, Basel, Switzerland) with the specific primer and SYBR GREEN Premix (Toyobo, Japan). Relative telomere length was calculated using T/S values using the following formula: T/S = 2 − ΔCq, where ΔCq = mean CqT − mean CqS. For normalization, β-Actin was used as a housekeeping gene. Each measurement was repeated in triplicate, and five serially diluted control samples were included in each experiment. 

### 2.3. RNA Isolation and mRNA Expression Analysis

Then, we extracted RNA from tissues using the TRIzol reagent (Molecular Research Center Inc., Cincinnati, OH, USA). RNA quality was measured by NanoDrop 1000 (Thermo Scientific, Wilmington, Denmark). Each cDNA was synthesized from 2 μg of total RNA using M-MLV reverse transcriptase (Promega, Madison, WI, USA). Then, qPCR was performed with a specific primer [19]. The sequences of TZAP (Bioneer, Daejeon, Republic of Korea) are as follows: Forward ‘AAG GCC CTT AGA GGC TGA AG’ and reverse ‘GAC TCC CTC CTG GTC AGC AC’. Further examination was carried out as previously described [19].

### 2.4. The Cancer Genome Atlas (TCGA) Data Analysis

The TCGA database from cBioPortal was used for the clinical and prognostic significance of TZAP. Data about the mRNA expression of TZAP were downloaded from https://tcga-data.nci.nih.gov/tcga/on (accessed on 1 January 2022) [20]. Its clinicopathological and prognostic values were analyzed. 

### 2.5. Cell Culture and siRNA Transfection

Huh1, Huh7, PLC/PRF/5, and HepG2 cell lines were obtained from the Korean Cell Line Bank (KCLB; Seoul, Republic of Korea). RPMI-1640 medium (Corning Incorporated, Corning, NY, USA) with 10% fetal bovine serum (HyClone Laboratories, Logan, UT, USA) and 1% of penicillin/streptomycin solution (HyClone Laboratories) was used for cell maintenance. Before a transfection, cells were maintained at 30% density, and Lipofectamine 2000 (Invitrogen, Carlsbad, CA, USA) and Opti-MEM (Invitrogen) were added as described in a previous study [19]. TZAP-specific siRNA (si-TZAP) and negative control siRNA (si-NC) duplexes were designed and supplied by Ambion (Austin, TX, USA; siRNA ID# s6566, s6567, s6568). The sequences of si-NC are as follows: 5′-ACGUGACACGUUCGGAGAA-3′ (sense) and 5′-UUCUCCGAACGUGUCACGU-3′ (antisense).

### 2.6. Cell Viability Assay

Cell viability was examined by the MTT reagent according to the manufacturer’s protocol (Duchefa Biochemie, Haarlem, Netherlands). Log-phase cells were trypsinized into single-cell suspension, and cells (1 × 10^3^ cells) were seeded in 96-well plates. Cells were transfected after 24 h as described above. After 24 h, 90 μL of plain media and MTT were mixed and cultured for 2 h in a 37 °C humidified incubator. Then, 100 µL of dimethyl sulfoxide (DMSO, Sigma-Aldrich, St. Louis, MO, USA) was mixed and OD was measured using an Asys UVM 340 microplate reader (Biochrom, Cambridge, UK).

### 2.7. Statistical Analysis

Chi-square and Student’s *t*-tests were performed for the analysis of the associations between the variables. Survival analysis was performed using the log-rank test as the univariate Kaplan–Meier curve. Overall survival (OS) and disease-free survival (DFS) were the times from diagnosis to mortality and disease recurrence or distant metastasis, respectively. A *p* value of <0.05 denotes significance for each statistical analysis.

## 3. Results

### 3.1. TZAP Expression in HCC Patients

TZAP expression was analyzed in 72 HCCs, and its expression level was similar in HCC and non-tumorous tissues (*p* = 0.67). When its level was standardized by the tumor/non-tumor ratio, the average expression level of TZAP was 1.81 ± 0.83. To identify the clinicopathological significance of TZAP, patients were divided into two groups according to its average values. Therefore, higher TZAP expression was shown in 29 (40.3%) of HCC 72 patients. High TZAP expression tended to be associated with the AST value (*p* = 0.222) and T stage (*p* = 0.136); however, it did not reach statistical significance. Additionally, other clinicopathological variables did not have any association with the TZAP expressions (Table 1).

Then, we assessed survival analysis to examine the prognostic value of TZAP expression in HCC. The median follow-up period was 66.39 months (3–105 months). Overall survival analysis showed a shorter survival in HCC patients with higher TZAP expression (53.14 vs. 73.95 months, χ2 = 5.43, *p* = 0.020) (Figure 1). Higher TZAP expression had an association with shorter disease-free survival (27.72 vs. 49.25 months, χ2 = 6.32, *p* = 0.012).

### 3.2. TZAP Expression in the Cancer Genome Atlas (TCGA) Data 

Clinicopathological characteristics of TZAP mRNA expression were analyzed from TCGA data and presented in Table 2. A total of 360 HCC patients were divided into two groups based on their median values. TZAP mRNA expression was higher in HCC with hepatitis C and lower in HCC with hepatitis B (*p* = 0.023). Serum AFP level was higher in HCC patients with higher TZAP expression; however, it did not have statistical significance (*p* = 0.171). 

Survival analysis revealed that TZAP expression did not have prognostic value in HCC (Figure 2). Overall survival results were similar in the two groups (1862.74 vs. 2035.87 days, χ2 = 0.304, *p* = 0.581). Disease-free survival analysis showed a shorter survival in HCC patients with lower TZAP expression; however, it was not significant (1703.39 vs. 1368.24 days, χ2 = 0.77, *p* = 0.38).

### 3.3. Cell Viability by TZAP Silencing

To confirm the effect of TZAP on cancer cell survival, cells were transfected with TZAP siRNA or a negative control. Then, we showed the efficacy of TZAP siRNA by qPCR. TZAP expression was statistically lower in cells transfected with TZAP siRNA than in cells transfected with si-NC (Figure 3a). The cell viability of two groups was examined by the MTT assay. In HepG2 cells, cell viability in the siRNA-treated group was approximately 140% higher than in the control group. However, cell viability was reduced by about 80% in PLC/PRF/5 cells (Figure 3b). Cell viability in Huh1 and Huh7 cell lines was unaffected by TZAP status.

## 4. Discussion

For the first time, we demonstrated the clinical value of TZAP expression in HCC. The newly identified TZAP may have a crucial role in telomere length and cancer development [9,13]. TZAP mutations have been known to have a poor prognosis in breast cancer; however, they are extremely rare in cancer [21]. In view of the role of TZAP in cancer cells, its higher expression may induce telomere shortening; therefore, we consider this protein a cancer suppressor [13]. In the TCGA data, patients with higher TZAP expression had better overall survival, as expected [22]. Other studies also showed that higher TZAP expression predicted a poorer survival result in colon cancer [19]. Moreover, TCGA and cell line results were different in the same study. Furthermore, the viability of TZAP-siRNA-treated cells was found to be reversed depending on the cell line. To clarify the role of TZAP in cancer, we studied the characteristics of TZAP expression in HCC tissues from patients, TCGA data, and HCC cell lines. 

Compared to normal tissue, TZAP mRNA expression was increased in HCC tissue; however, it was not significant. We previously studied TZAP expression in other cancers, such as colorectal cancer [19], cervical cancer [22], and lung cancers [23], using TCGA data or patient tissues. In these cancers, TZAP expression levels were slightly increased compared to normal tissues. A previous study also showed TZAP mRNA levels in all kinds of cancers using TCGA data [13]. They discovered that TZAP is significantly downregulated in kidney chromophobes and significantly upregulated in esophageal cancer, head and neck cancer, renal clear cell carcinoma, and HCC. In this study, there was no statistical difference between normal and HCC tissues. This disparity could be attributed to the different pathogenesis of HCC patients. Most Korean patients with HCC were related to the hepatitis B virus, though it was mostly associated with the hepatitis C virus in the western population [16]. It should be further confirmed in a larger patients’ group.

Additionally, TZAP mRNA expression did not show any clinical characteristics in HCC tissues. According to the TCGA data, TZAP expression was, interestingly, related to HCC with hepatitis C infection and without B hepatitis infection. According to one study, the TERT promoter mutation is strongly linked to hepatitis C and hepatitis B infection in HCC. Interestingly, TZAP and TERT expression on TCGA data were positively correlated in various cancers, including HCC [24]. In our HCC patients, TZAP expression tended to be associated with the T stage. However, it was not significant because of the low case number of the T4 stage of HCC. Considering the important roles of TZAP and TERT in telomere regulation and their positive correlation, hepatitis infection status may induce sudden changes in telomere regulation in the liver. 

In survival analysis, we found a poorer survival result in HCC patients with higher TZAP expression. It was contrary to the previous hypothesis that high expression of TZAP may induce telomere shortening and that it was associated with cancer cell death and a better prognosis [9,13,19]. Lower TZAP expression was associated with a poor prognosis in lung adenocarcinomas (OS and DFS), pancreatic adenocarcinomas (OS and DFS), cervical squamous cell carcinomas (OS), endocervical adenocarcinomas (OS), kidney renal papillary cell carcinomas (OS), and kidney renal clear cell carcinomas (DFS), according to TCGA data. In contrast, higher TZAP was associated with poorer prognosis in adrenocortical carcinoma (OS and DFS), brain lower grade glioma (OS and DFS), colon adenocarcinoma (OS), and prostate adenocarcinoma (DFS) [13]. Additionally, there was a limited amount of detail about TZAP expression in each cancer. In lung adenocarcinoma, lower TZAP expression was associated with a higher T stage, inducing a poorer prognosis [23]. Additionally, a shorter overall survival was found in cervical cancer with a lower TZAP expression, especially at the N1 stage [22]. In colorectal cancer, TZAP expression showed a paradoxical effect on TERT expression [19]. In patients’ tissues with CRC, lower TZAP expression and shorter telomeres had a poorer prognosis. However, TCGA data showed that lower TZAP expression was associated with a better prognosis in colon cancer. Moreover, TZAP silencing induced cancer cell growth in HCT116 cells; however, it decreased the viability of HT29 cells. They hypothesized that short telomeres uncontrolled by TZAP were important for cancer progression, and that TZAP expression was linked to TERT expression [19]. 

A similar result was found in this study. In patients’ tissues with HCC, higher TZAP expression had a poorer prognosis. Moreover, TZAP silencing increased cell growth in HepG2 cells, which was similar to the result from HCC patients. However, cell viability was decreased in the PLC/PRF/5 cell line. In other cell lines, TZAP status did not affect cell viability. Furthermore, according to the TCGA big data, TZAP expression in HCC had no prognostic value. It showed an inconsistent status of TZAP expression in HCC according to the data types. This disparity may have resulted from different HCC pathogenesis. Additionally, unknown mechanisms of telomere regulation may be present. We also investigated a correlation between TZAP and TERT in HCC patients’ tissues; however, it did not have any significance (data not shown). Therefore, not only TZAP expression but also other telomere-related genetic changes should be studied comprehensively. TZAP competes with the shelter-in-proteins TRF1 and TRF2 to bind telomeric DNA [9]. However, previous studies showed a positive correlation between TZAP and TRF1 and TRF2, suggesting that the balance of these expressions may be essential for telomere homeostasis [13]. We should re-identify the function of these telomeric regulation genes in light of their association with TZAP and other genes. Then, its molecular mechanism should be identified, and the prognostic value of TZAP in various cancers should be confirmed.

## 5. Conclusions

Here, we revealed the clinical significance of TZAP mRNA expression in HCC tissues, public data, and cell lines. We found that TZAP expression may have diverse effects on hepatocellular carcinogenesis. Our results warrant future study about the molecular mechanisms of telomere regulation via TZAP expression in HCC. 

## Figures and Tables

**Figure 1 medicina-58-01778-f001:**
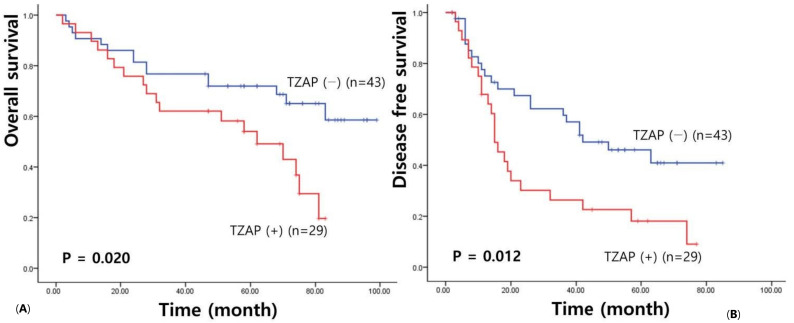
TZAP expression survival analysis in HCCs: (**A**) Overall survival; and (**B**) disease-free survival.

**Figure 2 medicina-58-01778-f002:**
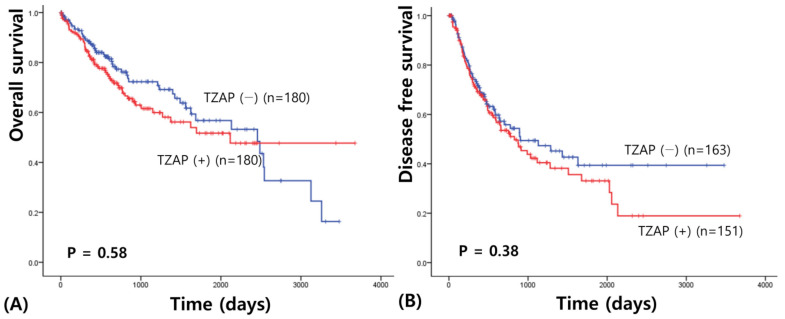
TCGA data about the prognostic value of TZAP mRNA expression in HCC: (**A**) Overall survival of TZAP expression; and (**B**) disease-free survival.

**Figure 3 medicina-58-01778-f003:**
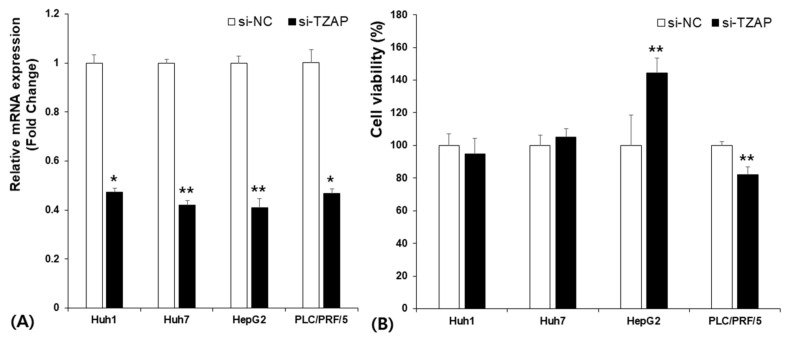
siRNA knockdown of TZAP and cell viability in HCC: (**A**) TZAP expression in Huh1, Huh7, HepG2, and PLC/PRF/5 cells after transfection; and (**B**) cell viability after transfection. * *p* < 0.05; ** *p* < 0.01.

**Table 1 medicina-58-01778-t001:** Clinicopathological significance of TZAP expressions in HCC.

	TZAP Expression		
	(+)	(−)	*p*
Total	29 (40.3)	43 (59.7)	
Age			0.461
<60	20 (43.5)	26 (56.5)	
≥60	9 (34.6)	17 (65.4)	
Gender			0.797
Male	23 (41.1)	33 (58.9)	
Female	6 (37.5)	10 (62.5)	
AST			0.222
<40	19 (36.5)	33 (63.5)	
≥40	10 (52.6)	9 (47.4)	
ALT			0.550
<40	21 (38.9)	33 (61.1)	
≥40	8 (47.1)	9 (52.9)	
Tumor size			0.292
<5 cm	16 (35.6)	29 (64.4)	
≥5 cm	13 (48.1)	14 (51.9)	
T stage			0.136
T1	5 (62.5)	3 (37.5)	
T2	16 (32.0)	34 (68.0)	
T3	7 (53.8)	6 (46.2)	
T4	1 (100)	0 (0)	
Telomere length			0.792
Short	18 (39.1)	28 (60.9)	
Long	11 (42.3)	15 (57.7)	

**Table 2 medicina-58-01778-t002:** TCGA data on TZAP expression in HCC.

	TZAP Expression (N, %)		*p*
	High (n = 180)	Low (n = 180)	
Age	60.23 ± 12.31	58.67 ± 14.19	0.62
Sex			0.73
Male	124 (50.6)	121 (49.4)	
Female	56 (48.7)	59 (51.3)	
T stage			0.26
T1	96 (53.9)	82 (46.1)	
T2	45 (50.6)	44 (49.4)	
T3	35 (44.3)	44 (55.7)	
T4	4 (30.8)	9 (69.2)	
N stage			1.00
N0	114 (46.2)	133 (53.8)	
N1	2 (50.0)	2 (50.0)	
M stage			1.00
M0	125 (47.9)	136 (52.1)	
M1	1 (33.3)	2 (66.7)	
Stage			0.50
I	91 (54.2)	77 (45.8)	
II	41 (49.4)	42 (50.6)	
III	37 (44.0)	47 (56.0)	
IV	2 (50.0)	2 (50.0)	
Serum AFP (µg/L)			0.171
≥20	59 (46.5)	68 (53.5)	
<20	81 (54.7)	67 (45.3)	
Child-Pugh			0.26
A	108 (50.0)	108 (50.0)	
B	7 (35.0)	13 (65.0)	
C	1 (100)	0 (0)	
Risk factor			0.023
Hepatitis B	26 (35.6)	47 (64.4)	
Hepatitis C	20 (64.5)	11 (35.5)	
Alcohol	37 (55.2)	30 (44.8)	
Others	97 (51.3)	92 (48.7)	

## Data Availability

Not applicable.

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
