# Peer review of "Clinical Characteristics of TZAP (ZBTB48) in Hepatocellular Carcinomas from Tissue, Cell Line, and TCGA"

_medicina, 2022, doi:10.3390/medicina58121778_

Round 1

Reviewer 1 Report

The authors want to demonstrate the clinical-pathological of TZAP in HCC. There is not enough evidence to prove it in this manuscript.

1.  In Line 20, "However, TZAP expression has not been investigated in cancers". In fact, TZAP expression showed in lung cancer, colorectal cancer, and cervical cancer.

2. The sequence of TZAP siRNA and internal control from qPCR were lacked in material and method, 

3. In Line 133, 1 × 103 cells?

4. In Line 134, "After 96 h, plain media and MTT were mixed and cultured...", The 96h for the cell transfect siRNA was too long, because the project of transfection need free FBS. These cells were starvation.

5.  siRNA-induced gene knockdown has inevitable off-target effects. The author should use two or more siRNAs to verify the off-target effects. 

Author Response

Thank you for your kind review.

----------------------------------------

The authors want to demonstrate the clinical-pathological of TZAP in HCC. There is not enough evidence to prove it in this manuscript.

  1. In Line 20, "However, TZAP expression has not been investigated in cancers". In fact, TZAP expression showed in lung cancer, colorectal cancer, and cervical cancer. --> It was revised

2. The sequence of TZAP siRNA and internal control from qPCR were lacked in material and method, --> It was revised and added. 

3. In Line 133, 1 × 103 cells? --> It was error and revised. 

4. In Line 134, "After 96 h, plain media and MTT were mixed and cultured...", The 96h for the cell transfect siRNA was too long, because the project of transfection need free FBS. These cells were starvation. --> It was error and revised. 

5.  siRNA-induced gene knockdown has inevitable off-target effects. The author should use two or more siRNAs to verify the off-target effects. 

-> We performed bigdata and patients data analysis additionally. Therefore, most effective siRNA was used. To clarify molecular mechanism of TZAP, further study will be performed using two or more siRNAs. 

Reviewer 2 Report

In this manuscript, the authors evaluated the clinical significance of TZAP (ZBTB48) in hepatocellular carcinomas from patients tissue, cell line, and TCGA. 

According to Dos Santos GA, et al., TZAP was either down-regulated or up-regulated depending on the tumor type. TZAP expression is associated with a favorable prognosis and is associated with better overall and disease-free survival. Looking at specific tumors, TZAP expression has a dual action. Its down-regulation is associated with poor prognosis in cervical squamous cell carcinoma, renal renal clear cell carcinoma, renal papillary cell carcinoma, lung adenocarcinoma and pancreatic adenocarcinoma. Conversely, upregulation of TZAP in adrenocortical carcinoma, colon and rectal cancer, brain low-grade glioma, and prostate adenocarcinoma is associated with poor prognosis.  

They have analysed 72 HCC of patients with resection, The level of TZAP expression was not associated with clinical parameters. TZAP expression was related poorer survival. TCGA data showed TZAP expression was more frequently in HCCs with hepatitis C infection, did not predict HCC prognosis. In cell line study, TZAP inhibition via siRNA were evaluated.

 Major point

To avoid selection bias, it is necessary to provide statistical evidence of the number of patients in the group with respect to the study design. The authors need to demonstrate statistical assumptions about sample size. Otherwise, please configure the cohort by presenting a flow chart for the target period.  

Minor point

Please index the indicated abbreviations in the manuscript.

(Reference)

Dos Santos GA, et al. Telomeric zinc-finger associated protein (TZAP) in cancer biology: friend or foe? Mol Biol Res Commun. 2021 Sep;10(3):121-129.

Author Response

Thank you for your kind review.

------------------------------

Major point

To avoid selection bias, it is necessary to provide statistical evidence of the number of patients in the group with respect to the study design. The authors need to demonstrate statistical assumptions about sample size. Otherwise, please configure the cohort by presenting a flow chart for the target period.  

--> The description about patients sample size were added in Material and method part. 

Minor point

Please index the indicated abbreviations in the manuscript.

--> The abbreviations were checked. 

(Reference)

Dos Santos GA, et al. Telomeric zinc-finger associated protein (TZAP) in cancer biology: friend or foe? Mol Biol Res Commun. 2021 Sep;10(3):121-129.

--> This reference was already cited in this manuscript. 

Round 2

Reviewer 1 Report

All questions had been solved in this revise.

Author Response

Thank you for your kind review.

Reviewer 2 Report

Raised issues are well resolved. 

(Minor)

Page  2, line 82 : 31 samples had poor quality~. -> Thirty-one samples were of poor quality and 59 samples were excluded from the study due to insufficient amount of RNA.

Author Response

Thank you for your kind suggestion. 

Page  2, line 82 : 31 samples had poor quality~. -> Thirty-one samples were of poor quality and 59 samples were excluded from the study due to insufficient amount of RNA.

--> It is better. We revised it as your suggestion.